# The Platelet Collagen Receptor GPVI Is Cleaved by Tspan15/ADAM10 and Tspan33/ADAM10 Molecular Scissors

**DOI:** 10.3390/ijms23052440

**Published:** 2022-02-23

**Authors:** Chek Ziu Koo, Alexandra L. Matthews, Neale Harrison, Justyna Szyroka, Bernhard Nieswandt, Elizabeth E. Gardiner, Natalie S. Poulter, Michael G. Tomlinson

**Affiliations:** 1School of Biosciences, University of Birmingham, Birmingham B15 2TT, UK; cxk543@bham.ac.uk (C.Z.K.); allymatthews33@googlemail.com (A.L.M.); neale.harrison@celentyx.com (N.H.); justynaszyroka@gmail.com (J.S.); 2Centre of Membrane Proteins and Receptors (COMPARE), Universities of Birmingham and Nottingham, Midlands B15 2TT, UK; n.poulter@bham.ac.uk; 3Institute of Experimental Biomedicine I, University Hospital and Rudolf Virchow Center Würzburg, University of Würzburg, D-97080 Würzburg, Germany; bernhard.nieswandt@virchow.uni-wuerzburg.de; 4Division of Genome Science and Cancer, John Curtin School of Medical Research, Australian National University, Canberra ACT 2601, Australia; elizabeth.gardiner@anu.edu.au; 5Institute of Cardiovascular Sciences, University of Birmingham, Birmingham B15 2TT, UK

**Keywords:** ADAM10, GPVI, tetraspanin, platelet, shedding, TspanC8, metalloproteinase

## Abstract

The platelet-activating collagen receptor GPVI represents the focus of clinical trials as an antiplatelet target for arterial thrombosis, and soluble GPVI is a plasma biomarker for several human diseases. A disintegrin and metalloproteinase 10 (ADAM10) acts as a ‘molecular scissor’ that cleaves the extracellular region from GPVI and many other substrates. ADAM10 interacts with six regulatory tetraspanin membrane proteins, Tspan5, Tspan10, Tspan14, Tspan15, Tspan17 and Tspan33, which are collectively termed the TspanC8s. These are emerging as regulators of ADAM10 substrate specificity. Human platelets express Tspan14, Tspan15 and Tspan33, but which of these regulates GPVI cleavage remains unknown. To address this, CRISPR/Cas9 knockout human cell lines were generated to show that Tspan15 and Tspan33 enact compensatory roles in GPVI cleavage, with Tspan15 bearing the more important role. To investigate this mechanism, a series of Tspan15 and GPVI mutant expression constructs were designed. The Tspan15 extracellular region was found to be critical in promoting GPVI cleavage, and appeared to achieve this by enabling ADAM10 to access the cleavage site at a particular distance above the membrane. These findings bear implications for the regulation of cleavage of other ADAM10 substrates, and provide new insights into post-translational regulation of the clinically relevant GPVI protein.

## 1. Introduction

Platelets are anucleate blood cells that are essential for haemostasis. At sites of vascular injury, subendothelial matrix proteins such as collagen are exposed to induce platelet activation, aggregation, thrombus formation and prevention of excessive blood loss. However, platelet thrombus formation elicited by rupture of an atherosclerotic plaque can give rise to occlusive arterial thrombosis which can result in myocardial infarction and ischemic stroke [1]. In addition to their established roles in haemostasis and thrombosis, platelets were recently shown to play pivotal roles in inflammation and cancer [2,3]. GPVI is the major platelet-activating receptor for collagen [4]. GPVI can also bind other ligands such as fibrin, fibrinogen, laminins, histones, diesel exhaust particles and other charged/hydrophobic ligands, although the physiological relevance of some of these ligands remains unclear [4]. Signal transduction following ligand binding to GPVI is initiated by phosphorylation of immunoreceptor tyrosine-based activation motifs (ITAMs) in the Fc receptor γ-chain (FcR γ-chain) dimer, which is constitutively associated with GPVI [4]. Despite well-characterized roles for GPVI in platelet activation, loss of this receptor in humans and mice does not cause major defects in haemostasis [5]. However, GPVI is important in arterial thrombosis and ongoing clinical trials are investigating blockade of the GPVI interaction with collagen as an antiplatelet strategy to prevent arterial thrombosis [6]. Moreover, GPVI is a key factor in maintaining vascular integrity during inflammation, and may play an additional role in venous thrombosis [5].

A disintegrin and metalloproteinase 10 (ADAM10) is a ubiquitously expressed cell membrane-localized metalloproteinase, which cleaves the extracellular regions from its transmembrane protein substrates [7]. ADAM10 is the critical scissor and activator for Notch cell fate regulatory proteins, since genetic ablation of mouse ADAM10 is embryonically lethal due to impaired Notch signalling. ADAM10 has at least 100 other substrates including amyloid precursor protein and cadherins [7]. Moreover, ADAM10 is considered as the predominant sheddase for GPVI [8,9], although the related ADAM17 can also cleave GPVI in mouse platelets in response to certain agonists [8]. GPVI cleavage is induced by the GPVI ligands collagen, collagen-related peptide [10] and fibrin [11], the thiol-alkylating agent and metalloproteinase activator N-ethylmaleimide (NEM) [9], active coagulation Factor X [12] and pathological shear [13], but GPVI clustering on immobilized collagen is protective against cleavage [14]. Released, soluble GPVI ectodomain is a plasma biomarker of platelet activation and is elevated in patients with coronary artery disease, ischemic stroke, rheumatoid arthritis, inflammatory bowel disease, sepsis and trauma-induced coagulopathy [11,15,16,17]. In addition, it is associated with bleeding risks in heparin-induced thrombocytopenia [18].

ADAM10 was recently proposed to exist as six distinct scissors with different substrates, depending on which of the six regulatory tetraspanins it forms a complex with [7,19,20,21]. The tetraspanins are a superfamily of transmembrane proteins with four transmembrane helices, two extracellular regions and cytoplasmic N- and C- termini [7]. Tetraspanins are present in most multicellular organisms, including 33 members in humans, and affect a variety of cell functions by acting as membrane organizers. They bind to specific membrane protein partners and regulate their function by controlling their expression, trafficking, and lateral mobility and clustering in the membrane [7]. The six ADAM10-regulating tetraspanins are termed the TspanC8s as they bear eight structurally important cysteines within their main extracellular region; all other tetraspanins have 4, 6 or 7 cysteines [22,23]. The TspanC8s are Tspan5, Tspan10, Tspan14, Tspan15, Tspan17 and Tspan33. The existence of intimate TspanC8/ADAM10 complexes is consistent with their co-dependence for expression [24,25], the identification of ADAM10 as the principal binding partner for Tspan15, the functionality of forced Tspan15/ADAM10 heterodimers [25], and a conserved ADAM10 binding site on the Tspan15 extracellular region [26]. The identification of substrate specificities of individual TpsanC8/ADAM10 complexes is in its infancy, but notable discoveries show the importance of Tspan15 for cleavage of neuronal (N)-cadherin [26,27,28,29,30], Tspan5 and Tspan17 regulation of vascular endothelial (VE)-cadherin expression and promotion of T cell transmigration in an inflammatory model [31], and Tspan5 and Tspan14 for Notch activation [22,24,27,32,33,34]. The TspanC8(s) that promote GPVI cleavage are unknown, but human platelets express three TspanC8s, Tspan14, Tspan15 and Tspan33 [20]. Understanding GPVI regulation is important given the focus of current clinical trials on GPVI as an antiplatelet drug target [35,36], and the potential utility of GPVI as a plasma biomarker for platelet activation in various disease processes [37,38].

The current study aimed to identify which TspanC8(s) support GPVI cleavage and the method in which they achieve this. Compensatory roles were identified for Tspan15 and Tspan33 in promoting GPVI cleavage, but Tspan14 was not involved. Tspan15 was the most efficient in promoting GPVI cleavage. Mechanistically, Tspan15 appeared to be important in allowing ADAM10 to access the GPVI cleavage site at a particular distance above the membrane.

## 2. Results

### 2.1. GPVI Cleavage Is Dependent on Tspan15 and Tspan33 in Transfected HEK-293T Cells

Human platelets express three TspanC8s, namely Tspan14, Tspan15 and Tspan33 [20]. To investigate which of these are important for GPVI cleavage, knockouts for each were generated using CRISPR/Cas9 in the HEK-293T cell line, as such an experiment cannot be performed in human platelets. HEK-293T cells were chosen since they express all six TspanC8s at the mRNA level [39]. HEK-293T cells do not express GPVI, but cell surface expression is achieved by GPVI transfection and its cleavage detected by Western blotting [28]. Moreover, GPVI cleavage is specific to ADAM10 in HEK-293T cells since cleavage is abrogated in ADAM10-knockout cells and unaffected by knockout of ADAM17, the closest relative of ADAM10 (Appendix A). Therefore, wild-type and TspanC8-knockout cells were co-transfected with GPVI containing a cytoplasmic Myc tag, together with the GPVI-associated FcRγ chain, and GPVI cleavage was assessed by anti-Myc Western blotting. GPVI cleavage was reduced by more than 50% in Tspan15-knockout cells, but not in Tspan14-knockout and Tspan33-knockout cells (Figure 1A). This result was observed in untreated cells and in cells treated with the alkylating agent and metalloproteinase activator NEM. Since GPVI cleavage was abolished in ADAM10-knockout cells (Figure 1A), this suggested partial compensation by other TspanC8s in Tspan15-knockout cells. Tspan33 was a more probable candidate than Tspan14, since Tspan14 overexpression was previously shown to suppress GPVI cleavage [28]. Indeed, combined knockout of Tspan15 and Tspan33 abolished GPVI cleavage to the same extent observed in ADAM10-knockout cells (Figure 1A), suggesting compensatory roles for the two TspanC8s.

It is well established that TspanC8s are required for ADAM10 trafficking to the cell surface [22,23,29]. Therefore, it was important to assess whether the loss of GPVI cleavage observed in Tspan15/33 double knockout cells was simply due to the loss of ADAM10 from the cell surface. Using flow cytometry, ADAM10 surface levels were found to be reduced by 20%, 60%, 40% and 70% in Tspan14-knockout, Tspan15-knockout, Tspan33-knockout, and Tspan15/33 double knockout HEK-293T cells, respectively (Figure 1B). Thus, different surface levels of ADAM10 did not explain the GPVI cleavage data, since substantial surface ADAM10 remained present on Tspan15/33 double knockout cells, presumably in complex with Tspan14 and other TspanC8s. Taken together, these data demonstrate that GPVI cleavage is mediated by Tspan15/ADAM10 and Tspan33/ADAM10 scissors in HEK-293T cells.

### 2.2. Tspan15 and Tspan33 Are Required for Cleavage of Endogenous GPVI in HEL Cells

To extend on the previous findings to endogenously expressed GPVI, the HEL 92.1.7 cell line, which expresses Tspan14, Tspan15 and Tspan33, as well as Tspan5 and Tspan17 [39], was used. A panel of CRISPR/Cas9 knockout HEL cells were generated and treated with phorbol 12-myristate 13-acetate (PMA) for 72 h to increase GPVI expression. Cells were then treated with NEM and GPVI cleavage measured by Western blotting with antibodies to the extracellular and cytoplasmic regions of GPVI. GPVI cleavage was reduced by 80% and 90% in Tspan15-knockout and Tspan33-knockout cells, respectively, but was normal in Tspan14-knockout cells (Figure 2A). Combined knockout of Tspan15 and Tspan33 abolished GPVI cleavage, comparable to ADAM10-knockout cells (Figure 2B). NEM stimulation was required to detect GPVI cleavage in these experiments (Figure 2A, B). The specificity of the two GPVI antibodies was confirmed by generation of GPVI-knockout HEL cells (Figure 2A and data not shown).

To determine whether the GPVI cleavage data in HEL cells was merely a consequence of Tspan15/33-knockout effects on ADAM10 expression, surface ADAM10 levels were measured using flow cytometry. Analysis revealed reductions of only 40%, 30%, 30% and 50% in Tspan14-knockout, Tspan15-knockout, Tspan33-knockout, and Tspan15/33 double knockout HEL cells, respectively (Figure 2C). These relatively subtle effects on ADAM10 expression fail to explain the striking GPVI cleavage data. Taken together, these data show that Tspan15/ADAM10 and Tspan33/ADAM10 act as the scissors for endogenous GPVI in HEL cells.

### 2.3. Tspan15/ADAM10 Is the Most Efficient Scissor for GPVI in HEK-293T Cells

To investigate the mechanism underlying the capacity of Tspan15 and Tspan33 to enable GPVI cleavage by ADAM10, an experiment was first designed to definitively determine which is the most potent. To achieve this, Tspan15/33 double knockout HEK-293T cells were transfected with FLAG-tagged Tspan15 and Tspan33 expression constructs to assess the extent of GPVI cleavage rescue with similar expression levels of the tetraspanins. Tspan15 expression restored some GPVI cleavage under basal conditions, and this was dramatically increased by NEM treatment (Figure 3). In contrast, Tspan33 expression only minimally restored GPVI cleavage following NEM treatment, despite comparable expression levels with Tspan15 (Figure 3). As a control, Tspan14 did not restore GPVI cleavage (Figure 3). These data suggest that Tspan15/ADAM10 is the most efficient scissor for GPVI. Tspan15 was therefore selected as the focus for follow-up mechanistic studies.

### 2.4. The Extracellular Region of Tspan15 Is Required for Efficient GPVI Cleavage and the C-terminus Is Inhibitory

To identify which regions of Tspan15 are important for its promotion of GPVI cleavage, a panel of six Tspan15 mutants and Tspan15/14 chimeric constructs were generated (Figure 4A). These were transfected into Tspan14/15/33 triple knockout HEK-293T cells, to prevent interference from endogenous expression of these tetraspanins. The mutant constructs were co-transfected with GPVI, FcRγ and ADAM10, the latter to ensure that ADAM10 was not limiting, since its surface expression was reduced by 90% in the triple knockout cells (data not shown). As expected, co-transfection of wild-type Tspan15 with ADAM10 dramatically restored GPVI cleavage in the triple knockout cells, significantly beyond what was observed in mock-transfected wild-type cells (Figure 4B). Four of the mutant/chimeric constructs behaved similarly to wild-type Tspan15 in this assay: a chimera with Tspan15 extracellular and Tspan14 transmembrane/intracellular regions, 14(15EC); a chimera with Tspan15 extracellular/transmembrane and Tspan14 intracellular, 15(14Cyto); a Tspan15 mutant lacking the intracellular tails, 15(ΔNC); and a chimera that comprised Tspan15 possessing a Tspan14 C-terminal tail, 15(14C) (Figure 4B). The common denominator among these constructs was the Tspan15 extracellular region, thus demonstrating its importance. Two of the constructs failed to restore GPVI cleavage: a chimera with Tspan14 extracellular and Tspan15 transmembrane/intracellular regions, 15(14EC); and a chimera comprised of Tspan14 possessing the Tspan15 C-terminal tail, 14(15C) (Figure 4B). The failure of these two chimeras to restore GPVI cleavage was particularly striking given that co-transfected wild-type Tspan14 with ADAM10 restored some GPVI cleavage in this assay, albeit significantly lower than observed with Tspan15 (Figure 4B). Importantly, all mutants and chimeras promoted ADAM10 maturation (Appendix A), suggesting they are correctly folded and functional. Taken together, these data reinforce the importance of the Tspan15 extracellular region for promoting GPVI cleavage, and highlight the Tspan15 C-terminal tail as a region with an apparent inhibitory or interfering effect.

### 2.5. The Capacity of Tspan15 Wild-Type and Mutant Forms to Promote GPVI Cleavage Is Unrelated to the Extent to Which They Co-Localize with GPVI

To determine whether accessibility to GPVI could explain the GPVI cleaving capacities of the Tspan15 mutant and chimeric constructs studied in Figure 4, their degree of co-localization with GPVI in Tspan14/15/33 triple knockout HEK-293T cells was investigated by Airyscan confocal microscopy. This advanced imaging technology allows lateral resolution of 120 nm and an enhanced signal-to-noise ratio compared to standard confocal microscopy [40]. The transfected cells were stained for Myc and FLAG to image GPVI and tetraspanin, respectively (Figure 5(Ai)). GPVI was localized predominantly at the cell surface. The strongest co-localization with GPVI was exhibited by wild-type Tspan15 and a chimera with Tspan15 extracellular and Tspan14 transmembrane/intracellular regions, 14(15EC), followed by a chimera with Tspan14 extracellular and Tspan15 transmembrane/cytoplasmic regions, 15(14EC), and wild-type Tspan14 (Figure 5(Aii)). The remainder of the mutants showed substantially less co-localization with GPVI (Figure 5(Aii)). When combined with the GPVI cleavage data from Figure 4, there was no correlation with the degree of co-localization (Figure 5B). This suggests that specificity of Tspan15 and mutant forms in promoting GPVI cleavage cannot be explained by accessibility to GPVI.

To understand whether this observation could be supported through imaging of endogenously expressed proteins, GPVI co-localization with ADAM10 was investigated in Tspan14-knockout and Tspan15/33 double knockout HEL cells. ADAM10 was imaged as a proxy for the TspanC8s since effective antibodies are not available to Tspan14 or Tspan33. Thus, if Tspan15/33 are promoting GPVI cleavage by co-localising ADAM10 with GPVI, then minimal co-localization of ADAM10 and GPVI should be observed in Tspan15/33 double knockout cells, but the reverse would be the case in Tspan14-knockout cells. However, there was no significant difference in the degree of GPVI and ADAM10 co-localization between the two cell types (Figure 6). These findings support the idea that preferential cleavage of GPVI by Tspan15/ADAM10 and Tspan33/ADAM10 scissors cannot be explained by any specific co-localization with GPVI.

### 2.6. Evidence That Cut Site Position on GPVI Contributes to Specific Cleavage by Tspan15/ADAM10 and Tspan33/ADAM10

TspanC8s have the potential to regulate ADAM10 by affecting its conformation, as different TspanC8s appear to interact with different regions on ADAM10 [28]. Additionally, there is evidence of association between ADAM10 cut site height above the membrane and the identity of the regulatory TspanC8 for particular substrates [7]. Specifically, N-cadherin [41] and betacellulin [42] contain cut sites positioned 10 and 7 residues above the predicted membrane interface, respectively, and both are cleaved by Tspan15/ADAM10 [25,26,27,28,29,30]. Notch1, the only known Tspan14/ADAM10 substrate, contains a cut site positioned at 15 amino acids above the membrane [43]. The ADAM10 cut site for GPVI is located 5 amino acids above the membrane [9]. These observations raise the possibility that different TspanC8s may position ADAM10 to preferentially cleave substrates with cut sites at particular positions above the membrane.

To investigate whether the scissor identity would change with cleavage site height, two GPVI mutants were generated in which the ADAM10 cleavage site was extended by five or ten residues above the membrane through insertion of glycine-serine linker sequences (Figure 7A). Since ADAM17 was found to cleave the stalk-extended mutants to a small extent in the absence of ADAM10 (Appendix A), ADAM10/17 double knockout HEK-293T cells were employed in these experiments to prevent interference from ADAM17. The cells were transfected with each of the six different TspanC8s and ADAM10, and GPVI cleavage measured by Western blotting. For wild-type GPVI, cleavage was significantly increased only when the true GPVI scissors, Tspan15/ADAM10 and Tspan33/ADAM10 were overexpressed, compared to overexpression of ADAM10 alone (Figure 7B). The magnitude of these increases were subtle, at only 2.2-fold and 1.8-fold, respectively, possibly because this is an overexpression system and endogenous TspanC8s are present. Indeed, ADAM10 overexpression alone yielded a substantial GPVI cleavage of 28% (Figure 7B). However, when the cleavage site was extended by 5 or 10 amino acids above the membrane, the preferential cleavage by Tspan15/ADAM10 and Tspan33/ADAM10 was lost (Figure 7C,D). These data support the hypothesis that different TspanC8/ADAM10 complexes cleave substrates that bear cleavage sites at particular positions above the membrane.

## 3. Discussion

This study has demonstrated that tetraspanins Tspan15 and Tspan33 bear redundant roles in promoting GPVI cleavage by ADAM10, but Tspan15 is the dominant ADAM10 regulator between the two. The underlying mechanism does not appear to be due to any specific capacity of these tetraspanins to co-localize ADAM10 in proximity with GPVI at a subcellular level. Instead, mechanistic experiments show the Tspan15 extracellular region to be critical and suggest that Tspan15 facilitates ADAM10 to access a cleavage site on GPVI at a particular distance above the membrane.

The importance of Tspan15 and Tspan33 in promoting GPVI cleavage was discovered using a panel of CRISPR/Cas9 knockout human cell lines for each the three TspanC8s reported to be expressed by platelets in a proteomic study, namely Tspan14, Tspan15 and Tspan33 [20]. Cell lines were required for this study as genetic modification of human platelets in vitro is not possible. The use of cell lines represented a limitation of this study, but importantly, the capacity of Tspan15 and Tspan33 to promote GPVI cleavage was verified in two different cell lines. These were HEK-293T cells that do not express GPVI, thus GPVI was transfected into the cells with its associated FcRγ signalling chain, and HEL cells that express the GPVI/FcRγ complex endogenously. In both cell lines, the dual knockout of Tspan15 and Tspan33 was required to abolish GPVI cleavage, equivalent to knockout of ADAM10. In contrast, Tspan14 knockout had no effect. TspanC8s promote ADAM10 trafficking to the plasma membrane [22,23,29], but the absence of GPVI cleavage in Tspan15/33 double knockout cells was not due to a generalized loss of ADAM10 expression from the cell surface. This was because ADAM10 surface expression was only partially reduced following Tspan15/33 combination knockout, by 70% and 50% in HEK-293T and HEL cells. This suggests that in these cells, the remaining ADAM10 complexes with Tspan14 and other TspanC8s are unable to cleave GPVI. One difference observed between the cell lines was that single knockouts of Tspan15 and Tspan33 both significantly reduced GPVI cleavage in HEL cells, whereas in HEK-293T cells, Tspan15 knockout reduced cleavage of GPVI but loss of Tspan33 had no effect. The likely reasons underlying this are a relatively high expression level of Tspan33 in HEL cells versus HEK-293T cells [39], and the fact that Tspan15 is more efficient than Tspan33 in promoting GPVI cleavage. The latter was demonstrated by the observation that when similar levels of Tspan15 and Tspan33 were transfected into Tspan15/33 double knockout HEK-293T cells, restoration of GPVI cleavage by Tspan15 was substantially greater than by Tspan33. On human platelets, a proteomic study estimated copy numbers of 2000, 2500 and 2100 molecules per platelet for Tspan14, Tspan15 and Tspan33, respectively [20]. These numbers require validation when antibodies become available to all three of these TspanC8s. Nevertheless, the comparable expression levels of Tspan15 and Tspan33, together with Tspan15′s greater capacity to promote GPVI cleavage, suggest that Tspan15/ADAM10 is the dominant scissor for GPVI on human platelets.

To investigate how Tspan15/33 promotes GPVI cleavage, Tspan15 was selected for mechanistic studies as the most efficient in this regard. A panel of six Tspan15 mutant and Tspan14/15 chimeric constructs were generated to investigate which regions of Tspan15 are important for promoting GPVI cleavage. The extracellular region of Tspan15 was found to be critical since all mutants/chimeras that included this region supported efficient GPVI cleavage. This is consistent with a previous study showing that the extracellular regions of TspanC8s are important for interaction with ADAM10 [28]. In contrast, the cytoplasmic tails were not required since mutants which lacked the cytoplasmic tails, but retained the rest of Tspan15, could still promote GPVI cleavage to the same extent as wild-type Tspan15. A potential negative regulatory role for the C-terminal tail of Tspan15 was identified using two chimeras consisting mostly of Tspan14 but with the C-terminal tail of Tspan15 in common. These entirely failed to support GPVI cleavage, even though wild-type Tspan14 supported some GPVI cleavage when co-expressed with ADAM10. In wild-type Tspan15, the positive effect of the extracellular region clearly overrode the inhibitory effect of the C-terminal tail. How the latter exerts its inhibitory effect remains unclear, but it may recruit a cytoplasmic regulatory protein. As a precedence for this, the Tspan33 C-terminal tail recruits pleckstrin homology domain-containing family A member 7 (PLEKHA7) to promote ADAM10 localization to adherens junctions of polarized epithelial cells [44].

To investigate whether Tspan15 promotes GPVI cleavage by regulating ADAM10 subcellular localization and drawing the metalloproteinase into closer proximity with its substrate, fluorescence microscopy was used to assess the degree of co-localization of the six Tspan15 mutant/chimeric constructs with GPVI. Co-localization in HEK-293T cells did not correlate with shedding capacity. Furthermore, in HEL cells, endogenous ADAM10 and GPVI co-localization was similar in Tspan15/33 double knockout versus Tspan14-knockout cells, which have opposite cleavage phenotypes. These findings suggest Tspan15 does not promote GPVI shedding by co-localising ADAM10 with its substrate. This does not rule out the possibility that TspanC8s mediate some interaction between TspanC8/ADAM10 complexes and substrate. However, such interactions are likely to be relatively low affinity and transient, as TspanC8 and ADAM10 substrate interactions have not been readily identified in mass spectrometry proteomic studies [25,27,45].

The discoveries outlined in the previous paragraph led to a new hypothesis that Tspan15 promotes GPVI cleavage by enabling ADAM10 to cleave at a particular distance above the membrane. The rationale for this was two-fold. First, different TspanC8s were previously shown to bind to distinct regions on ADAM10, suggesting that each may cause ADAM10 to adopt a different conformation [28]. Second, there is an association between the TspanC8 that promotes cleavage of a particular substrate, with the distance of ADAM10 cleavage site above the membrane, estimated by the number of amino acid residues [7]. For example, the GPVI cleavage site is located 5 amino acids above the membrane [9], somewhat similar to N-cadherin [41] and betacellulin [42] at 10 and 7 residues above the membrane, respectively, and cleavage of all are promoted by Tspan15 [25,26,27,28,29,30]. In contrast, Notch1 contains a cleavage site positioned at 15 amino acids above the membrane [43], and cleavage is promoted by Tspan5 and Tspan14 [22,24,27,32,33,34]. In agreement with the hypothesis, extending the ADAM10 cleavage site on GPVI by 5 or 10 amino acids abolished its dependence on Tspan15 and Tspan33. These data have implications for the general mechanism by which TspanC8s regulate ADAM10 specificity, suggesting that TspanC8s interact with ADAM10 in such a way as to direct ADAM10 to cut sites at particular distances above the membrane. In the case of Tspan15 and Tspan33, this appears to be relatively close at 5–10 amino acids from the membrane. A limitation of these mechanistic experiments is the reliance on mutant constructs transfected into HEK-293T cells. In future, cryo-electron microscopy analyses of each TspanC8/ADAM10 complex will determine whether ADAM10 is positioned differently in each.

In conclusion, a major finding of this study is the identification of Tspan15/ADAM10 and Tspan33/ADAM10 as molecular scissors for GPVI, a clinically relevant antiplatelet target for myocardial infarction and ischemic stroke. Theoretically, a therapeutic strategy designed to activate Tspan15/ADAM10 or Tspan33/ADAM10 may provide a novel treatment for arterial thrombosis, since irreversible shedding of GPVI would render platelets non-responsive to collagen. However, specific activation of a particular TspanC8/ADAM10 complex may be difficult to achieve and may induce undesirable side effects on other substrates and/or other cell types. Additional major findings of this study are the novel insights into how TspanC8s regulate ADAM10 substrate specificity. At least part of the mechanism appears to involve different TspanC8/ADAM10 complexes preferentially cleaving substrates that contain cleavage sites at particular distances above the membrane.

## 4. Materials and Methods

### 4.1. Antibodies

Primary antibodies were mouse anti-Myc (9B11) (Cell Signaling Technology), rabbit anti-FLAG (F7425) (Merck), mouse anti-HA (6E2) (Cell Signaling Technology), rabbit anti-human GPVI cytoplasmic tail [46], mouse anti-human GPVI (11A7) [47], mouse anti-human GPVI (336A9) (Bernhard Nieswandt, unpublished), mouse anti-human ADAM10 (11G2) (a gift from Eric Rubinstein, Paris, France) [47] and control mouse IgG1 (MOPC-21) (MP Biomedicals).

### 4.2. Expression Constructs

The human GPVI expression construct in the pcDNA3.1 vector and containing a Myc epitope tag at the C-terminus, and the human untagged FcRγ-chain expression construct in the pEF6/Myc-His vector, were as described [48]. HA-tagged ADAM10 in the pcDNA3.1 vector was as described [49]. The GPVI stalk extension mutants were generated by inserting sequences encoding GGGGS residues between the extracellular and transmembrane region of wild-type GPVI, using the Q5 site-directed mutagenesis kit (New England Biolabs, Hitchin, UK). TspanC8s in the pEF6/Myc-His-FLAG vector had been described previously [23]. Synthesized fragments of extracellular domain chimeras between Tspan14 and Tspan15 (Thermo Fisher Scientific, Horsham, UK) and Tspan15 with the cytoplasmic domain replaced by that of Tspan14 (Twist Bioscience, South San Francisco, CA, USA) were subcloned into the pEF6/Myc-His-FLAG vector. Tspan15 lacking N- and C-terminal tails, and chimeras between Tspan14 and Tspan15 at the C-terminus, were generated by PCR cloning using wild-type human Tspan14 and Tspan15 as templates.

### 4.3. CRISPR Guide RNA Design

Knockout cell lines were generated by CRISPR/Cas9 as described [25]. Guide RNAs were selected using the Wellcome Trust Sanger Institute CRISPR design tool [50] based on proximity to the translation start site and the fewest possible off-target effects. The guides for ADAM10 and Tspan15 were described previously [25]. For the other genes, the primer pairs containing the flanking BbsI sites for annealing and cloning into the pSpCas9(BB)-2A-Puro plasmid (a gift from Feng Zhang, Addgene plasmid #62988) were as follows:ADAM17 guide 1: 5′ CACCGCCGAAGCCCGGGTCATCCGG 3′ and 5′ AAACCCGGATGACCCGGGCTTCGGC 3′ADAM17 guide 2: 5′ CACCGCGAAAGGAACCACGCTGGTC 3′ and 5′ AAACGACCAGCGTGGTTCCTTTCGC 3′Tspan14 guide 1: 5′ CACCGTTATAGATACTCTAACGCCA 3′ and 5′ AAACTGGCGTTAGAGTATCTATAAC 3′Tspan14 guide 2: 5′ CACCGCAGATCGATATCGTCCCGGT 3′ and 5′ AAACACCGGGACGATATCGATCTGC 3′Tspan33 guide 1: 5′ CACCGGCTCGGCTAATGAAGCATGC 3′ and 5′ AAACGCATGCTTCATTAGCCGAGCC 3′Tspan33 guide 2: 5′ CACCGGAAGGAGAACTCCTCCCCGG 3′ and 5′ AAACCCGGGGAGGAGTTCTCCTTCC 3′GPVI guide: 5′ CACCGCCGACCGCCCTCTTCTGTCT 3′ and 5′ AAACAGACAGAAGAGGGCGGTCGGC 3′

Knockout clones were verified by flow cytometry for Tspan15, ADAM10, ADAM17 and GPVI, and by genomic sequencing for Tspan14 and Tspan33.

### 4.4. Cell Culture and Transfections

The human embryonic kidney (HEK)-293T (HEK-293 cells expressing the large T-antigen of simian virus 40) cell line was cultured in complete Dulbecco’s Modified Eagle’s medium (cDMEM) (Merck, Gillingham, UK) containing 10% fetal bovine serum (Thermo Fisher Scientific), 4 mM L-glutamine, 100 units/mL penicillin and 100 μg/mL streptomycin (Merck). The human erythroleukemia (HEL) 92.1.7 cell line was cultured in complete Roswell Park Memorial Institute-1640 (cRPMI) (Merck) containing the same supplements as cDMEM. To increase GPVI expression levels, HEL cells were activated with 6.2 ng/mL phorbol 12-myristate 13-acetate (PMA) (Merck) for 72 h. Transient transfections in HEK-293T cells were carried out using polyethylenimine (Merck) as described [51,52]. HEL cells were transfected by electroporation at 280 V, 950 μF using a method described previously for other cell types [48].

### 4.5. GPVI Cleavage Assay

For GPVI cleavage experiments in HEK-293T cells, cells were transfected with expression constructs for human GPVI, incorporating a C-terminal myc tag, and FcRγ. In some experiments, 2 mM NEM (Thermo Fisher Scientific) was added to activate metalloproteinases for 30 min prior to harvesting the cells for anti-Myc Western blotting. The percentage of GPVI cleavage was calculated by expressing the lower, cleaved GPVI band as a percentage of total GPVI (full-length and cleaved fragment bands). For endogenous GPVI cleavage experiments in PMA-treated HEL cells, 2 mM NEM was added for 1 h before harvesting the cells for Western blotting. Relative GPVI cleaved fragment was calculated by normalising the cleaved fragment, and detected using rabbit anti-human GPVI (cytoplasmic tail), to full-length GPVI, which was measured using mouse anti-human GPVI antibody (11A7). Relative GPVI cleaved fragment data were made relative to the NEM-treated wild-type sample, which was arbitrarily set at 100.

### 4.6. Western Blotting

All cell lysates were prepared using Triton X-100 lysis buffer (1% Triton X-100, 10 mM Tris pH 7.5, 150 mM NaCl, 1 mM EDTA, 0.02% NaN_3_) (Merck). Western blotting was performed as previously described [28] using an Odyssey Infrared Imaging System and IRDye 680RD- or 800CW-conjugated secondary antibodies (LI-COR Biosciences, Cambridge, UK).

### 4.7. Flow Cytometry

For flow cytometry of ADAM10 on HEK-293T, cells were stained with 10 μg/mL allophycocyanin (APC)-conjugated mouse anti-human ADAM10, or the equivalent negative control mouse IgG (R&D Systems, Abingdon, UK). For flow cytometry of ADAM10 on HEL, cells were stained with 10 μg/mL mouse anti-human ADAM10 or isotype control mouse IgG1, followed by goat anti-mouse IgG-APC (Thermo Fisher Scientific). Samples were processed on a FACSCalibur flow cytometer and data were collected and analyzed using CellQuest Pro software (BD Biosciences, Wokingham, UK). The level of ADAM10 surface expression on each cell type was then calculated by dividing the geometric mean fluorescence intensity of the ADAM10 staining by the isotype control value.

### 4.8. Confocal Microscopy

Cells were fixed, permeabilized and immunostained as previously described [25]. Secondary antibodies were Alexa Fluor 488- or 647-conjugated (Thermo Fisher Scientific). For simultaneous labelling of ADAM10 and GPVI, cells were first stained with 2.5 μg/mL GPVI mAb (336A9), then Alexa Fluor 488-conjugated goat anti-mouse secondary antibody, and finally 1 μg/mL of Alexa Fluor-647-conjugated ADAM10 (11G2) mAb. Dual-colour confocal images were captured sequentially with 488 nm and 640 nm laser lines using a 63 × 1.4 NA oil objective in super-resolution mode on a Zeiss LSM 900 microscope with an Airyscan 2 detector (Carl Zeiss). The images were post-processed on ZEN software (Carl Zeiss) to obtain Airyscan images. The degree of co-localization was analyzed as described previously [25].

### 4.9. Statistics

Relative or percentage data were normalized using arcsine transformation of the square root prior to statistical analysis with ANOVA. Post hoc multiple comparison tests were used to analyze differences between multiple groups; these are indicated in the figure legends together with significance values. All statistical tests were performed using GraphPad Prism software.

## Figures and Tables

**Figure 1 ijms-23-02440-f001:**
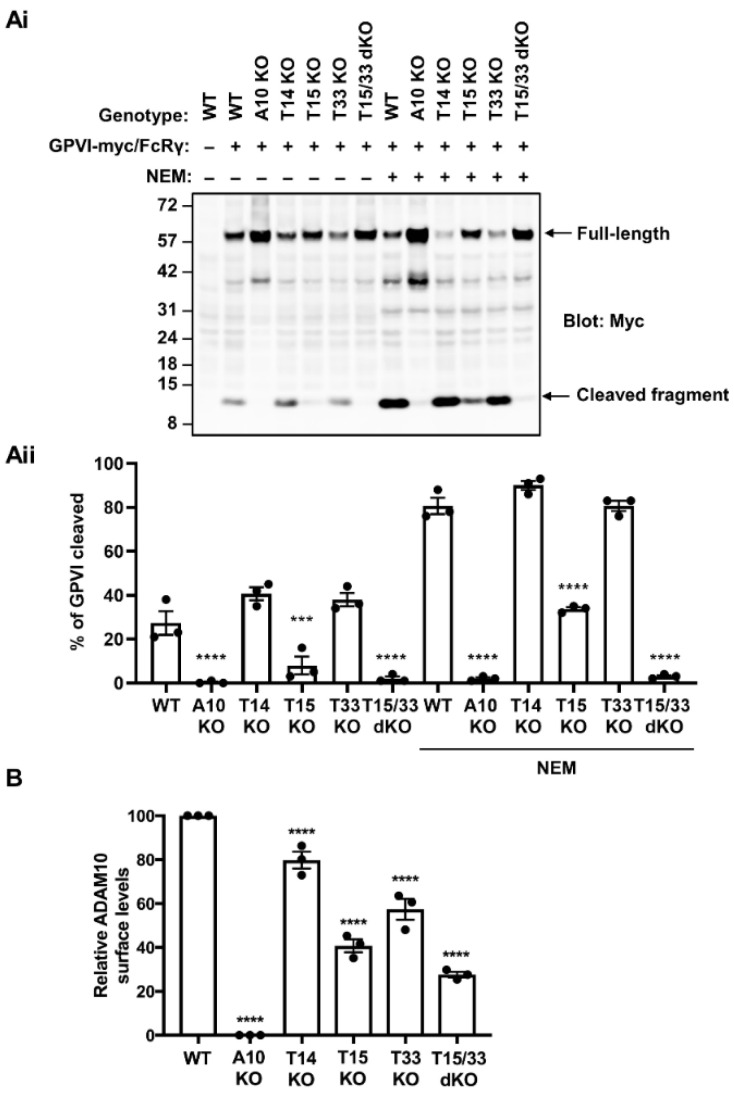
GPVI cleavage is dependent on Tspan15 and Tspan33 in transfected HEK-293T cells. (**Ai**) Wild-type (WT), ADAM10-knockout (A10 KO), Tspan14-knockout (T14 KO), Tspan15-knockout (T15 KO), Tspan33-knockout (T33 KO) and Tspan15/33 double knockout (T15/33 dKO) HEK-293T cells were transfected with expression constructs for C-terminally Myc-tagged human GPVI and FcRγ (+), or empty vector (–). Cells were treated with 2 mM NEM (+), or vehicle control (–) for 30 min prior to harvest. Cells were then lysed in 1% Triton X-100 and lysates subjected to Western blotting with an anti-Myc antibody. (**Aii**) The percentage of cleaved GPVI from Ai was calculated. Error bars represent the standard error of the mean from three independent experiments. Data were arcsine-transformed and statistically analyzed using a two-way ANOVA followed by a Dunnett’s multiple comparison test, compared to their respective WT controls in each stimulation condition (***, *p* < 0.001; ****, *p* < 0.0001). (**B**) ADAM10 surface expression on the cell types described in panel A were analyzed by flow cytometry. Geometric mean fluorescence intensity was used to assess surface ADAM10 levels and data were made relative to WT values. Error bars represent the standard error of the mean from three independent experiments. Average geometric mean intensities were arcsine transformed and then statistically analyzed using a one-way ANOVA followed by a Dunnett’s multiple comparison test, compared to WT (****, *p* < 0.0001). The data shown are from single CRISPR/Cas9 knockout clones, but similar data were generated using a second set of clones generated using different guide RNA sequences (data not shown).

**Figure 2 ijms-23-02440-f002:**
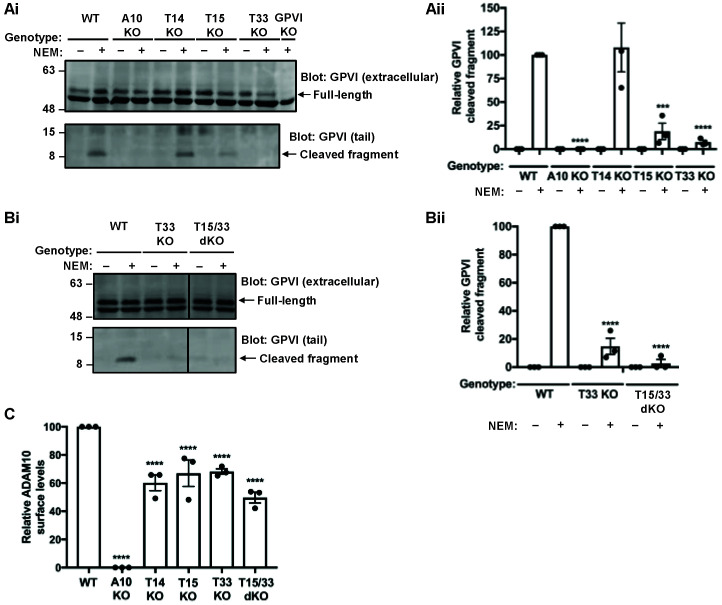
Tspan15 and Tspan33 are required for cleavage of endogenous GPVI in HEL cells. (**Ai**) Wild-type (WT), ADAM10-knockout (A10 KO), Tspan14-knockout (T14 KO), Tspan15-knockout (T15 KO), Tspan33-knockout (T33 KO) and GPVI-knockout (GPVI KO) HEL cells were treated with 6.2 ng/mL PMA for 72 h. Cells were stimulated with 2 mM NEM (+) for 1 h before harvest. Cells were lysed in 1% Triton X-100 and separated by SDS-PAGE. Lysates were probed with a mouse anti-human GPVI antibody (11A7), which recognises the extracellular region of GPVI and therefore only the full-length protein (top panel) and a rabbit anti-human GPVI antibody, which recognises the cytoplasmic tail of GPVI (bottom panel). (**Aii**) Relative GPVI cleavage was calculated by normalising the cleaved fragment signal to the full-length GPVI; the NEM-treated WT was arbitrarily set at 100. Error bars represent the standard error of the mean from three independent experiments. Data were normalised by arcsine transformation and statistically analyzed using a two-way ANOVA followed by a Bonferroni multiple comparison test, compared to WT (***, *p* < 0.001; ****, *p* < 0.0001). (**Bi**) WT, T33 KO and Tspan15/33 double knockout (T15/33 dKO) HEL cells were treated with PMA and subjected to lysis and Western blotting as described in panel (**Ai**). (**Bii**) Relative GPVI cleaved was quantitated and statistically analyzed as described in panel (**Aii**). (**C**) ADAM10 surface expression on PMA-differentiated cells used in panels A and B were analyzed by flow cytometry and quantitated as described in Figure 1B. The data shown are from single CRISPR/Cas9 knockout clones, but similar data were generated using a second set of clones generated using different guide RNA sequences (data not shown).

**Figure 3 ijms-23-02440-f003:**
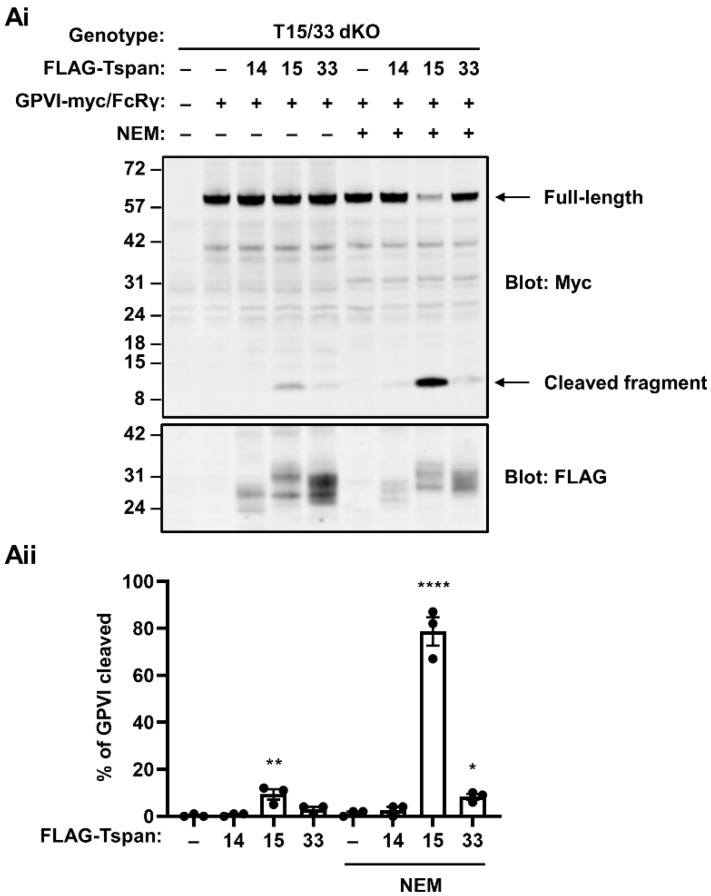
Tspan15/ADAM10 is the most efficient scissor for GPVI in HEK-293T cells. (**Ai**) Tspan15/33 double knockout (T15/33 dKO) HEK-293T cells were transfected with expression constructs for C-terminally Myc-tagged human GPVI and FcRγ (+) or empty vector (–), and FLAG-tagged human Tspan14, Tspan15, Tspan33 or empty vector control (–). Cells were treated with 2 mM NEM (+), or vehicle control (–) for 30 min prior to harvest, and then lysed in 1% Triton X-100 followed by anti-Myc and anti-FLAG Western blotting. (**Aii**) The percentage of cleaved GPVI from Ai was calculated. Error bars represent the standard error of the mean from three independent experiments. Data were arcsine-transformed and statistically analyzed using a two-way ANOVA followed by a Dunnett’s multiple comparison test, compared to their respective non-transfected controls in each stimulation condition (*, *p* < 0.05, **, *p* < 0.01; ****, *p* < 0.0001).

**Figure 4 ijms-23-02440-f004:**
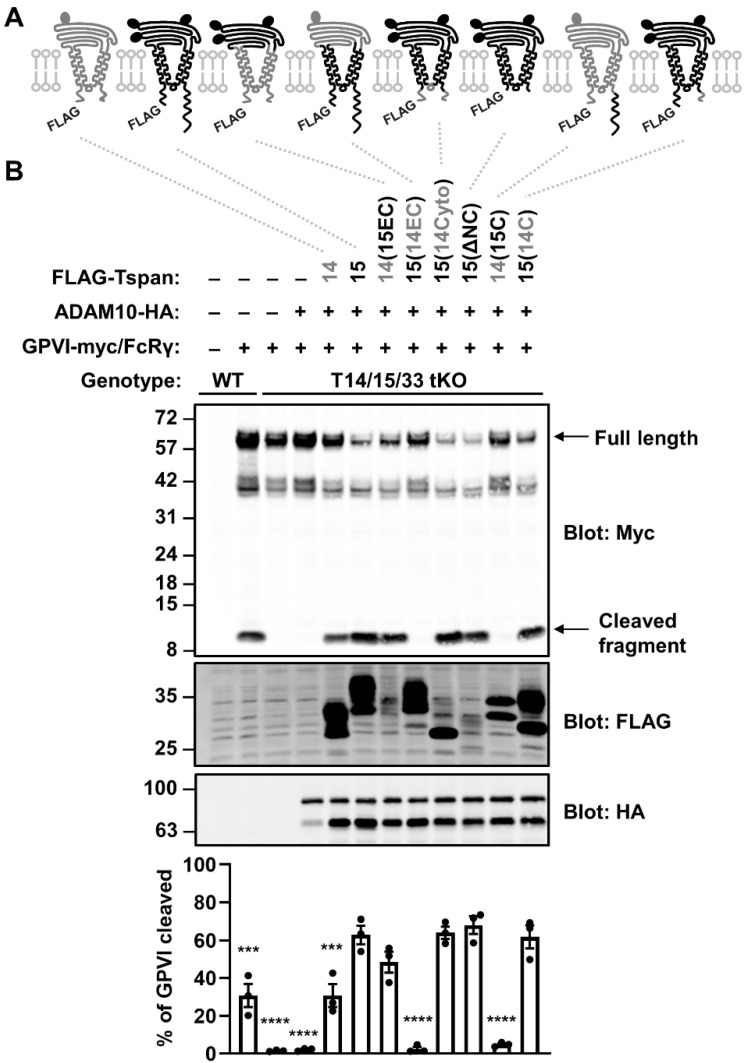
The extracellular region of Tspan15 is required for efficient GPVI cleavage and the C-terminus is inhibitory. (**A**) Schematic of N-terminally FLAG-tagged Tspan14 (grey) and Tspan15 (black) chimeras, where the entire extracellular region (EC), cytoplasmic domain (Cyto) or C-terminal tail (C) were exchanged. Truncation of both the N- and C-terminal tails is indicated by ΔNC. Ovals represent N-glycosylation sites. (**B**) Wild-type (WT) HEK-293T cells were transfected with expression constructs for C-terminally Myc-tagged GPVI and FcRγ (+) or empty vector (–). In addition to GPVI and FcRγ, Tspan14/15/33 triple knockout (T14/15/33 tKO) HEK-293T cells were co-transfected with HA-tagged ADAM10 alone, or in combination with FLAG-tagged Tspan14 and Tspan15 constructs described in panel A. Cells were lysed in 1% Triton X-100 followed by Western blotting with anti-Myc, anti-FLAG and anti-HA antibodies (top panels). The percentage of cleaved GPVI was calculated (lower panel). Error bars represent the standard error of the mean from three independent experiments. Data were arcsine-transformed and statistically analyzed using a one-way ANOVA followed by a Dunnett’s multiple comparison test, compared to T14/15/33 tKO cells transfected with wild-type Tspan15 and ADAM10 (***, *p* < 0.001; ****, *p* < 0.0001).

**Figure 5 ijms-23-02440-f005:**
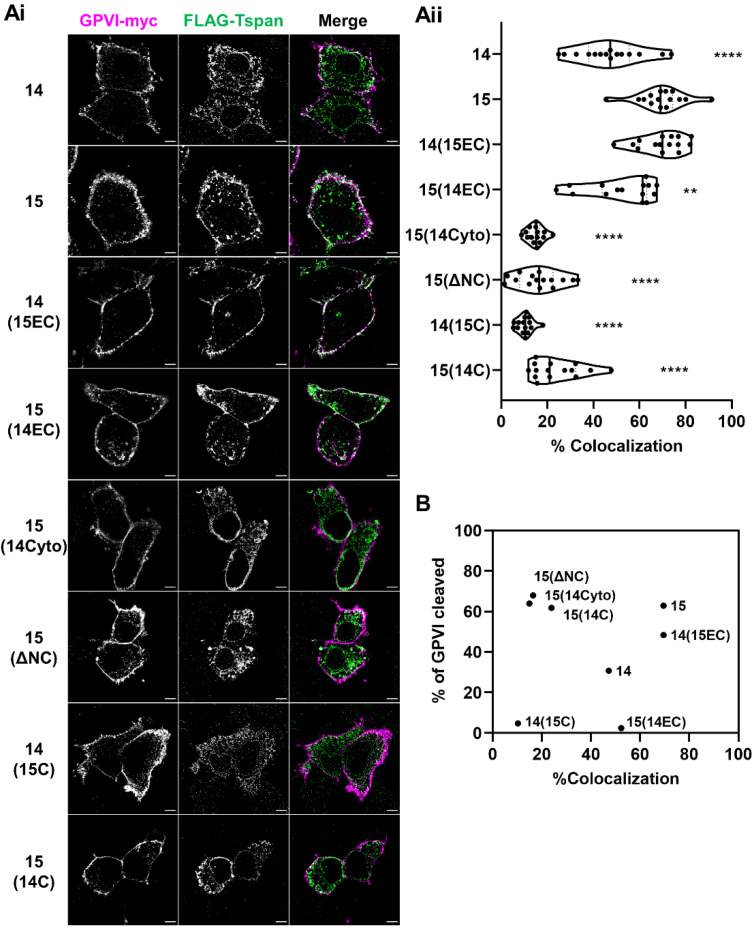
The capacity of Tspan15 wild-type and mutant forms to promote GPVI cleavage is unrelated to the extent to which they co-localise with GPVI. Tspan14/15/33 triple knockout HEK-293T cells were transfected with expression constructs for C-terminally Myc-tagged GPVI, FcRγ, ADAM10 and FLAG-tagged Tspan14 or 15 chimeras that are described in detail in Figure 4A. Cells were fixed and stained with anti-Myc (GPVI–Myc, magenta) and anti-FLAG (FLAG-Tspan, green) antibodies. (**Ai**) The middle planes of the cells were imaged using Airyscan confocal microscopy in super-resolution mode (scale bar 5 μm). No signal was detected in either channel in the empty vector-transfected cells (data not shown). (**Aii**) The degree of co-localization between GPVI–Myc and FLAG-TspanC8s is presented as the percentage of co-localizing pixels in the GPVI–Myc (magenta) channel. Data are representative of 15 fields of view from three independent experiments and are normalized by arcsine transformation before being statistically analyzed by a one-way ANOVA, followed by Dunnett’s multiple comparison tests, compared to cells transfected with wild-type Tspan15 (**, *p* < 0.01; ****, *p* < 0.0001). (**B**) Scatter plot summarising the relationship between degree of co-localization, expressed as the average of data presented in panel **Aii**, and percentage of GPVI cleaved, expressed as mean of the quantitation presented in Figure 4B.

**Figure 6 ijms-23-02440-f006:**
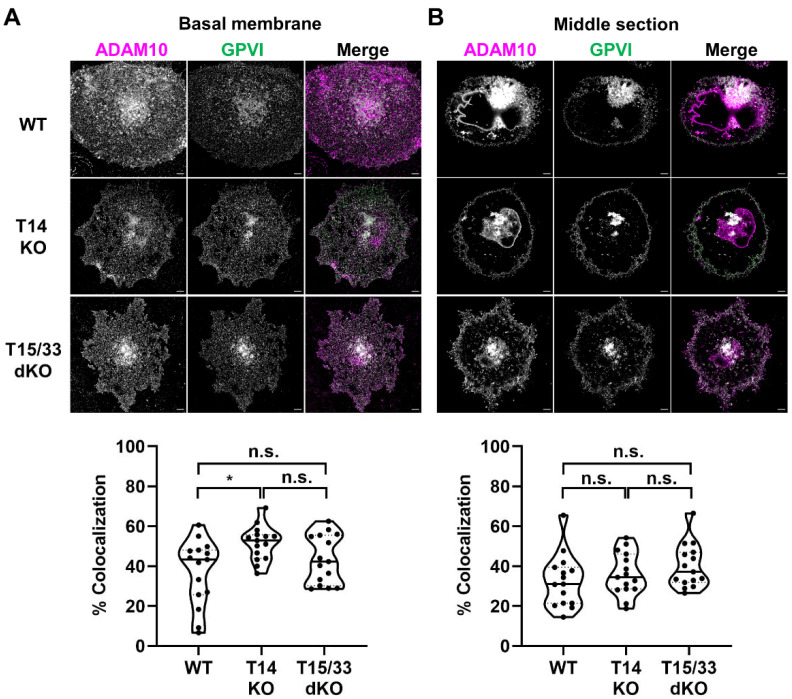
Co-localization between ADAM10 and GPVI is similar in Tspan14-knockout and Tspan15/33 double knockout HEL cells. Wild-type (WT), Tspan14-knockout (T14 KO) and Tspan15/33 double knockout (T15/33 dKO) HEL cells were treated with 6.2 ng/mL PMA for 72 h. Cells were fixed and stained with anti-ADAM10 mAb (11G2, magenta) and an antibody against the extracellular region of GPVI (336A9, green). Images of the (**A**) basal membrane and (**B**) middle section of the cells were captured using Airyscan confocal microscopy in super-resolution mode (top panels; scale bar 5 μm). No ADAM10 signal was detected in control ADAM10-knockout cells and no GPVI signal was detected in control GPVI-knockout cells (data not shown). The degree of co-localization between ADAM10 and GPVI is presented as the percentage of co-localizing pixels in the ADAM10 (magenta) channel (lower panels). Data are representative of 15 fields of view from three independent experiments and are normalized by arcsine transformation and statistically analyzed by a one-way ANOVA, followed by Bonferroni’s multiple comparison tests (n.s., not significant; *, *p* < 0.05).

**Figure 7 ijms-23-02440-f007:**
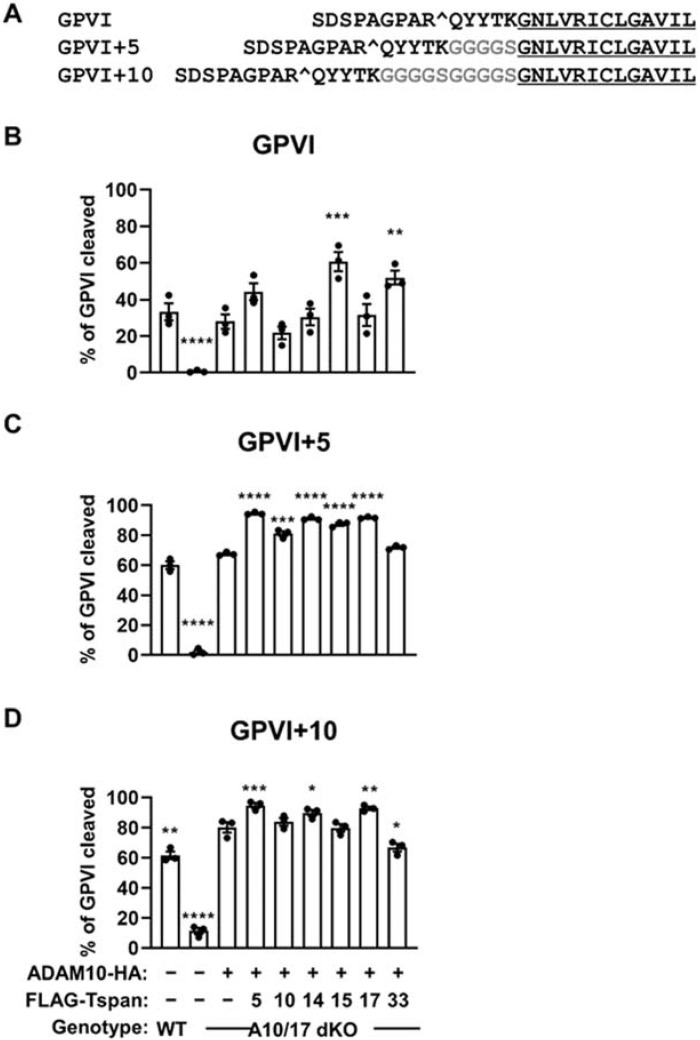
Evidence that cut site position on GPVI contributes to specific cleavage by Tspan15/ADAM10 and Tspan33/ADAM10. (**A**) Amino acid alignment of wild-type GPVI and the stalk extension mutants, GPVI+5 and GPVI+10: ADAM10 cut site is denoted by ^; residues inserted are highlighted in grey; transmembrane region is underlined. (**B**) Wild-type (WT) HEK-293T cells were transfected with expression constructs for C-terminally Myc-tagged GPVI and FcRγ (+) or empty vector (–). ADAM10/17 double knockout (A10/17 dKO) HEK-293T cells were co-transfected with GPVI and FcRγ, HA-tagged ADAM10 alone, or in combination with FLAG-tagged TspanC8s (Tspan5, 10, 14, 15, 17 or 33). Cells were lysed in 1% Triton X-100 followed by Western blotting with anti-Myc, anti-FLAG and anti-HA antibodies (data not shown). The percentage of cleaved GPVI was calculated. Cleavage assays for (**C**) GPVI+5 and (**D**) GPVI+10 was performed as described in panel B. Error bars represent the standard error of the mean from three independent experiments. Data were arcsine-transformed and statistically analyzed using a one-way ANOVA followed by a Dunnett’s multiple comparison test, compared to A10/17 dKO cells transfected with ADAM10 alone (*, *p* < 0.05; **, *p* < 0.01, ***, *p* < 0.001; ****, *p* < 0.0001).

## Data Availability

Not applicable.

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
