# Peer review of "The Platelet Collagen Receptor GPVI Is Cleaved by Tspan15/ADAM10 and Tspan33/ADAM10 Molecular Scissors"

_ijms, 2022, doi:10.3390/ijms23052440_

Round 1

Reviewer 1 Report

The aim of this manuscript is to evaluate which TspanC8 support GPVI cleavage and the mechanisms that lead to this process.

Even if the manuscript provides an organic overview, with a densely organized structure and based on well-synthetized evidence, there are aspects to be mentioned, to make the article fully readable. For these reasons, the manuscript requires minor changes.

Please find below an enumerated list of comments on my review of the manuscript:

INTRODUCTION:

LINE 35: Platelets represents a cellular subgroup of the elements, circulating in the bloodstream, which exert a pivotal role in responding to vessel injuries, regulating angiogenesis and the innate immunity, as reported by several and recent studies (see, for reference: Bianchi, S.; Torge, D.; Rinaldi, F.; Piattelli, M.; Bernardi, S.; Varvara, G. Platelets’ Role in Dentistry: From Oral Pathology to Regenerative Potential. Biomedicines 2022, 10, 218. https://doi.org/10.3390/biomedicines10020218).

LINE 70: With two extracellular regions and intracellular N – and C – termini, Tetraspanins are present in most multicellular organism, playing significant and pleiotropic tasks, as suggested by several and recent studies (see, for reference: Harrison, N.; Koo, C.Z.; Tomlinson, M.G. Regulation of ADAM10 by the TspanC8 Family of Tetraspanins and Their Therapeutic Potential. Int. J. Mol. Sci. 2021, 22, 6707. https://doi.org/10.3390/ijms22136707)

In conclusion, this manuscript is densely presented and well organized, based on well-synthetized evidences. The authors were lucid in their style of writing, making it easy to read and understand the message, portrayed in the manuscript. Besides, the methodology design was rigorous and appropriately implemented within the study. However, many of the topics are very concisely covered. This manuscript provided a comprehensive review of current knowledge in this field. Moreover, this research have futuristic importance and could be potential for future research. However, the minor concern of this manuscript is with the introductive section: for these reasons, I have minor comments only for the introductive section, for improvement before acceptance for publication. The article is accurate and provides relevant information on the topic and I suggest minor changes to be made in order to maximize its scientific impact. I would accept this manuscript, if the comments are addressed properly.

Author Response

We thank the reviewer for their positive comments on our manuscript and their two useful suggestions for improvement of the Introduction, which we think have improved this section.

The first suggestion was to provide a more detailed introduction to platelets and a new citation.  The new citation was a dentistry-focussed review on platelets.  To avoid any potential confusion that our manuscript might involve dentistry research in some way, we have instead chosen to include three new citations that are recent review articles focussing on the established role of platelets in haemostasis and thrombosis and the merging roles in immunity, inflammation, cancer and angiogenesis.  We have included the following additional sentence to accompany the new citations: “In addition to their established roles in haemostasis and thrombosis, platelets have recently been shown to play pivotal roles in inflammation and cancer”.

The second suggestion was to improve the general tetraspanin introduction and to include a citation for this, which we have done.

We hope the reviewer is satisfied with our response.

Reviewer 2 Report

I’ve read with attention the paper of Koo et al. that is potentially of interest. The background and aim of the study have been clearly defined. The methodology applied is overall correct, the results are reliable and adequately discussed. I’ve only some minor comments:

  • What Post-hoc multiple comparison tests have been applied? What p level has been chosen as significant? What software has been used to carry out statistical tests?
  • The Authors should shortly discuss the potential limitation of their research approach and the eventual next step of their research.

Author Response

We are grateful for the positive comments on the manuscript.  The reviewer had two comments for improvement and out response is outlined below.

Comment 1.  What post-hoc multiple comparison tests have been applied? What p level has been chosen as significant? What software has been used to carry out statistical tests?

Response: The post-hoc tests and p levels were indicated in the figure legends, and GraphPad Prism was used for the statistical tests.  To include the latter and clarify the former, we have added the following to the Methods section on Statistics: “(multiple comparison tests) are indicated in the figure legends together with significance values.  All statistical tests were done using GraphPad Prism software.”

Comment 2.  The Authors should shortly discuss the potential limitation of their research approach and the eventual next step of their research.

Response: We have addressed the limitations at two points in the Discussion.  Firstly, in the second paragraph, by adding, “The use of cell lines is a limitation of this study”, just after we explain that the experiments cannot be done in platelets.  Secondly, in the penultimate paragraph, by adding, “A limitation of these mechanistic experiments is the reliance on mutant constructs transfected into HEK-293T cells.  In future, cryo-electron microscopy analyses of the six TspanC8/ADAM10 complexes will help to determine whether ADAM10 adopts different positions in each.”

We hope that the reviewer shares our view that these additions have improved the manuscript.